# Upper Critical Solution Temperature (UCST) Behavior of Polystyrene-Based Polyampholytes in Aqueous Solution

**DOI:** 10.3390/polym11020265

**Published:** 2019-02-04

**Authors:** Komol Kanta Sharker, Yuki Ohara, Yusuke Shigeta, Shinji Ozoe, Shin-ichi Yusa

**Affiliations:** 1Department of Applied Chemistry, Graduate School of Engineering, University of Hyogo, 2167 Shosha, Himeji, Hyogo 671-2280, Japan; sharkerkomol@diu.edu.bd (K.K.S.); ery_5096@yahoo.co.jp (Y.O.); shinji.ozoe@tosoh-finechem.co.jp (S.O.); 2Tosoh Finechem Co., 4988 Kaisei-cho, Shunan, Yamaguchi 746-0006, Japan; yusuke.shigeta@tosoh-finechem.co.jp

**Keywords:** polyampholyte, UCST, RAFT, electrostatic interaction

## Abstract

Strong polyampholytes comprising cationic vinylbenzyl trimethylammonium chloride (VBTAC) bearing a pendant quaternary ammonium group and anionic sodium *p*-styrenesulfonate (NaSS) bearing a pendant sulfonate group were prepared via reversible addition-fragmentation chain-transfer polymerization. The resultant polymers are labelled P(VBTAC/NaSS)*_n_*, where *n* indicates the degree of polymerization (*n* = 20 or 97). The percentage VBTAC content in P(VBTAC/NaSS)*_n_* is always about 50 mol%, as revealed by ^1^H NMR measurements, meaning that P(VBTAC/NaSS)*_n_* is a close to stoichiometrically charge-neutralized polymer. Although P(VBTAC/NaSS)*_n_* cannot dissolve in pure water at room temperature, the addition of NaCl or heating solubilizes the polymers. Furthermore, P(VBTAC/NaSS)*_n_* exhibits upper critical solution temperature (UCST) behavior in aqueous NaCl solutions. The UCST is shifted to higher temperatures by increasing the polymer concentration and molecular weight, and by decreasing the NaCl concentration. The UCST behavior was measured ranging the polymer concentrations from 0.5 to 5.0 g/L.

## 1. Introduction

Stimuli-responsive polymers change their physical and chemical properties in response to changes in external conditions such as temperature, pH, solvent ionic strength, light irradiation, and the application of electric and magnetic fields. Among them, thermo-responsive polymers are the most widely studied because of potential application in fields such as drug release [1,2], gene therapy [3], bio-separation [4], thermally switchable optical devices [5], bioimaging [6], and catalysis [7]. Phase diagram analysis can differentiate thermo-responsive polymers into lower or upper critical solution temperature (LCST or UCST) types by the position at which the miscibility gap is observed, i.e., at high or low temperature, respectively [8,9]. Owing to these interesting features, LCST and UCST polymers have been studied by many research groups for more than five decades. In 1968, Heskins et al. [10] observed an LCST phase transition for poly(*N*-isopropylacrylamide) (PNIPAM) in water at around 32 °C, which is close to human body temperature. Since then, extensive research has been conducted into exploiting the thermo-responsive behavior of PNIPAM in biomedical and other fields. Furthermore, many other LCST polymers, such as poly(*N*-isopropylmethacrylamide) [11], poly(*N*-vinylcaprolactam) [12], and poly(oligo(ethylene glycol) (meth)acrylate) [13], have been prepared. However, far fewer UCST polymers in water have been reported. UCST polymers become soluble upon heating, and this quality has the potential to be exploited in biomedicine as a means for auto-regulated drug delivery in response to increasing body temperature. Recently, Agarwal et al. [14] among others [15,16,17] have studied hydrogen-bonding UCST polymers. However, zwitterionic polymers that bear both cationic and anionic charges on the same pendant chain are also good candidate UCST polymers. For example, poly(*N*-(3-sulfopropyl)-*N*-methacroyloxyethyl-*N*,*N*-dimethyl ammonium betaine) bears cationic ammonium and anionic sulfonate groups on the same pendant chain and exhibits UCST behavior in water owing to the strong interactions between the charged groups [18,19].

Polyampholytes are composed of cationic and anionic monomers. The charge interactions therein endow such polymers with special characteristics, making them promising for various applications [20,21,22,23,24,25,26]. However, very few studies in the literature have addressed the controlled radical polymerization of polyampholytes or identifying and maintaining the proper ratios of anionic and cationic monomers [27]. Controlling the chemical structures and accordingly the charge balances, molecular weights, and molecular weight distributions (*M*_w_/*M*_n_) of polyampholytes can be used to tailor their properties. For natural proteins, the proper ratio of anionic and cationic charges is important for their specific biological functions. Even a slight imbalance in these charges will lead to protein malfunction. Similarly, polyampholytes require structural control and the appropriate stoichiometric incorporation of monomer units to provide specific properties.

Zhang et al. [28] reported the preparation of amphoteric random copolymers from methacrylic acid and 2-(dimethylamino)ethyl methacrylate by reversible addition-fragmentation chain-transfer (RAFT) controlled radical copolymerization. The random copolymers exhibit UCST behavior in various alcohol/water solvent mixtures. However, the difference in the reactivities of the two monomers made it difficult to incorporate the monomers into the polymer chain at a suitable ratio and to maintain proper interactions between the charged groups. Cationic vinylbenzyl trimethylammonium chloride (VBTAC) and anionic sodium *p*-styrenesulfonate (NaSS) are well-known styrene-type monomers [24,29]. Control over the polymerization of a polyampholyte has a strong effect on its UCST behavior. For example, Agarwal et al. [30] did not observe UCST behavior in copolymers of styrene and acrylamide prepared via conventional free-radical polymerization. However, when prepared with RAFT polymerization, the resultant random copolymer exhibited UCST behavior.

Herein, we introduce strong polyampholytes prepared via RAFT that exhibit UCST behavior. These copolymers were prepared using VBTAC and NaSS monomers at two different degrees of polymerization (DP), i.e., P(VBTAC/NaSS)*_n_* where *n* indicates the DP and it is 20 or 97 in the present study (Scheme 1). P(VBTAC/NaSS)*_n_* exhibits UCST in aqueous NaCl solutions, which were characterized in terms of percentage transmittance (%*T*), hydrodynamic radius (*R*_h_), and light scattering intensity (SI) as well as by optical microscopy and fluorescence probe techniques. The UCST of this system increases with decreasing NaCl concentration ([NaCl]) and increasing polymer concentration (*C*_p_) and DP. Due to an H–D isotope effect, the UCST increases when deuterium oxide (D_2_O) is used as a solvent.

## 2. Experimental

### 2.1. Materials

Vinylbenzyl trimethylammonium chloride (VBTAC, 99%) from Sigma-Aldrich (St. Louis, MO, USA), 4,4′-azobis (4-cyanopentanoic acid) (V-501, 98%) from Wako Pure Chemical (Osaka, Japan), *N*-phenyl-1-naphthylamine (PNA, 98%) from Tokyo Chemical Industry (Tokyo, Japan), and poly(sodium *p*-styrenesulfonate) (PNaSS, *M*_w_ = 70,000) from Sigma-Aldrich were used as received without further purification. Sodium *p*-styrenesulfonate (NaSS, 98% estimated by high performance liquid chromatography) was purchased from Tokyo Chemical Industry and used as received without further purification. 4-Cyanopentanoic acid dithiobenzoate (CPD) was prepared according to a previously reported method [31]. Methanol (MeOH) was distilled after drying with 3 Å molecular sieves. Water was purified using an ion-exchange column system. All other reagents were used as received.

### 2.2. Preparation of Polyampholytes (P(VBTAC/NaSS)_n_)

First, we studied the relationship between polymerization time and conversion using equimolar amounts of VBTAC and NaSS via RAFT radical polymerization. VBTAC (212 mg, 1.00 mmol), NaSS (206 mg, 1.00 mmol), CPD (5.61 mg, 0.20 mmol), and V-501 (2.86 mg, 0.10 mmol) were dissolved in a mixed solvent of D_2_O (1.8 mL) containing 1.2 M NaCl and MeOH (0.2 mL) ([VBTAC]/[NaSS]/[CPD]/[V-501] = 50/50/1/0.5; molar ratio). The solution was transferred to an NMR tube. In-situ polymerization was performed at 70 °C under argon in an NMR apparatus in order to obtain NMR data at several time intervals. The conversion was estimated from the integral intensity of the vinyl proton signal observed at 5.7 ppm compared to that for the phenyl protons at 6–8 ppm.

P(VBTAC/NaSS)*_n_* (*n* = 20 and 97) were prepared using RAFT polymerization (Appendix A). P(VBTAC/NaSS)_20_ was prepared as a following method. VBTAC (530 mg, 2.51 mmol), NaSS (546 mg, 2.65 mmol), CPD (69.8 mg, 0.250 mmol), and V-501 (35.0 mg, 0.125 mmol) were dissolved in a mixed solvent of 1.2 M NaCl (4.50 mL) and MeOH (0.502 mL) ([VBTAC]/[NaSS]/[CPD]/[V-501] = 10/10/1/0.5; molar ratio). The solution was heated at 70 °C for 5 h under argon atmosphere. After polymerization, the total monomer conversion was found to be 99.2% as estimated using ^1^H NMR. The reaction mixture was dialyzed against 1.2 M NaCl for two days and then pure water for one day. P(VBTAC/NaSS)_20_ was recovered by freeze-drying (689 mg, 64.0%). P(VBTAC/NaSS)_97_ was also prepared by the same method (720 mg, 68.0%).

### 2.3. Measurements

^1^H NMR spectra were obtained with a Bruker DRX-500 spectrometer (Billerica, MA, USA) operating at 500 MHz. Infrared (IR) spectroscopy was performed on a Jasco (Tokyo, Japan) FT/IR-4200 by the attenuated total reflection (ATR) technique using an incident angle of 45°. The samples were analyzed over 256 scans. Jasco Spectra Manager Version 2 software was used to analyze the data. The phase separation temperatures of the aqueous polymer solutions were measured with respect to percent transmittance (%*T*) of a 700-nm light beam using a quartz sample cell with a 10-mm path length. %*T* was measured on a Jasco V-630BIO UV-vis spectrophotometer equipped with a temperature control system (Jasco ETC-717). The temperature was increased from 20 to 80 °C and then decreased from 80 to 20 °C at a heating and cooling rate of 1.0 °C/min. Dynamic light scattering (DLS) measurements were performed using a Malvern Zetasizer Nano ZS (Malvern, UK) equipped with a He-Ne laser (4 mW at 632.8 nm). The scattering angle (*θ*) was fixed 173° to measure light scattering intensity (SI). All sample solutions for DLS were filtered through a 0.2 µm membrane filter prior to analysis. The data obtained were analyzed using Zetasizer 7.11 software (Malvern, UK) to calculate hydrodynamic radius (*R*_h_) and polydispersity index (PDI). *R*_h_ and SI values used in this study are the averages of two measurements.

## 3. Results and Discussion

Strong polyampholytes (P(VBTAC/NaSS)*_n_*; *n* = 20 and 97) with different degrees of polymerization (DP = *n*) composed of cationic VBTAC and anionic NaSS were prepared via RAFT polymerization.

The relationship between polymerization time and total conversion (*p*) of VBTAC and NaSS in a mixed solvent of D_2_O containing 1.2 M NaCl and MeOH was studied (Figure 1). The polymerization was performed in NMR equipment at 70 °C under argon atmosphere. The *p* value was estimated from the vinyl peak at 5.7 ppm. We measured the total monomer conversion of VBTAC and NaSS, because the ^1^H NMR signals for the vinyl protons in VBTAC and NaSS completely overlap. An induction period of 4.5 min was observed in the initial stage of the polymerization, which is common for RAFT [32,33]. Sometimes induction period can be observed due to contaminations such as trace of oxygen, or impurities in monomers. After the induction period, the conversion increases with increasing polymerization time. The concentration of propagating radicals is constant from 4.5 to 100 min, as revealed by the linear first-order kinetic plot. Generally, alternating comonomer sequences seem preferred for the copolymerization of cationic with anionic monomers, independent of the polymerizable groups involved [20,34,35,36,37,38]. The theoretical degree of polymerization (DP(theory)) and number-average molecular weight (*M*_n_(theory)) were calculated using *p* and the following equations:(1)$$\mathrm{DP}\left(\mathrm{theory}\right)=\frac{{\left[M\right]}_{0}}{{\left[\mathrm{CTA}\right]}_{0}}\times \frac{p}{100}$$
(2)$$M_{n}\left(\mathrm{theory}\right)=\mathrm{DP}\left(\mathrm{theory}\right)\times M_{m}+M_{\mathrm{CTA}}$$
where [M]_0_ and [CTA]_0_ are the initial concentrations of the monomer and CTA, respectively, and *M*_m_ and *M*_CTA_ are the molecular weights of the monomer and CTA, respectively. The values of *M*_n_ (theory) for P(VBTAC/NaSS)_20_ and P(VBTAC/NaSS)_97_ are 4.18 × 10^3^ and 2.03 × 10^4^ g/mol, respectively (Table 1). Gel-permeation chromatography (GPC) measurements for P(VBTAC/NaSS)*_n_* could not be performed because the ampholytes cannot be dissolved in GPC eluents. ATR-IR spectra for P(VBTAC/NaSS)_20_ and P(VBTAC/NaSS)_97_ were obtained and found to be very similar (Appendix A). The characteristic peaks observed at 3033 and 2923 cm^−1^ are due to aromatic and aliphatic C–H stretching, respectively. The peaks at 1623 and 1482 cm^−1^ correspond to aromatic C=C stretching and alkyl C–H bending. The peak at 1183 cm^−1^ is due to –SO_3_^−^. Owing to the tendency of the monomers to absorb moisture from the environment, a further peak is observed at approximately 3400 cm^−1^.

Proton NMR measurements for P(VBTAC/NaSS)*_n_* were performed in D_2_O containing 1.2 M NaCl at 80 °C (Figure 2). In order to dissolve the polymers in solution, the temperature was raised to 80 °C, which is higher than the UCST. The main chain proton signals are observed at 0.8–2.3 ppm. The pendant phenyl proton signals are observed at 6.2–7.8 ppm. From the ratio of the integral intensities of the pendant aryl protons and the pendant methyl protons in the VBTAC units at 2.9 ppm, the VBTAC content in P(VBTAC/NaSS)*_n_* was estimated to be about 50 mol%. However, there is no evidence of the randomness of P(VBTAC/NaSS)*_n_*, because we cannot control the sequence of VBTAC and NaSS in the copolymerization. If VBTAC and NaSS homopolymers are mixed in D_2_O containing 1.2 M NaCl, we may obtain the same ^1^H NMR spectra in Figure 2. ^1^H NMR data merely indicated that P(VBTAC/NaSS)*_n_* contains about equimolar amounts of the VBTAC and NaSS units in the solution. The charges were nearly canceled in this solution. Indeed, zeta-potential of the aqueous solutions of P(VBTAC/NaSS)*_n_* were near to zero, which is to be discussed later. It is considered that we have obtained copolymers of VBTAC and NaSS, because both the monomers have styrene-type structures.

To measure phase transition temperature (*T*_p_) during heating and cooling processes, the temperature dependence of percentage transmittance (%*T*) for an aqueous P(VBTAC/NaSS)_20_ solution at [NaCl] = 0.1 M and *C*_p_ = 1.0 g/L was measured over two heating/cooling cycles (Appendix A). The tangent the %*T* line was extrapolated to %*T* = 100%, and the cross point was defined as *T*_p_. The values of *T*_p_ for the two heating processes are 54.5 and 51.5 °C, which is not consistent. During heating, precipitated polymers dissolve above the UCST. The polymer precipitates observed below the UCST are not homogeneous in terms of size and shape. *T*_p_ during heating depends on the size and shape of the polymer precipitates. Conversely, the values of *T*_p_ for the two cooling processes are the same (42.8 °C). During cooling, a homogeneous polymer solution is precipitated below the UCST. Thus, *T*_p_ is reproducible for the cooling process because the polymer chains precipitate from a homogeneous unimer state with decreasing temperature. Therefore, we focused on the cooling process to study the UCST behavior of the system.

To determine *T*_p_ values for aqueous P(VBTAC/NaSS)_20_ solutions with [NaCl] values from 0 to 0.2 M, the %*T* values were measured as a function of temperature at *C*_p_ = 2.0 g/L (Figure 3a). *T*_p_ shifts to lower temperature with increasing [NaCl] for the cooling process. The *T*_p_ values for aqueous P(VBTAC/NaSS)_20_ at [NaCl] = 0 and 0.2 M are 55.7 and 31.4 °C, respectively. The %*T* values of aqueous P(VBTAC/NaSS)_97_ solutions with different [NaCl] values from 0 to 2.0 M were measured as a function of temperature at *C*_p_ = 2.0 g/L (Figure 3b). The *T*_p_ values for P(VBTAC/NaSS)_97_ also shift to lower temperature with increasing [NaCl]. Although *T*_p_ for P(VBTAC/NaSS)_20_ in pure water is 55.7 °C, the %*T* value for P(VBTAC/NaSS)_97_ in pure water does not reach 100% until the temperature reaches 90 °C (Appendix A). In other words, P(VBTAC/NaSS)_97_ does not dissolve in pure water at any temperature. This observation indicates that attractive electrostatic interactions increase with increasing molecular weight. At the same [NaCl], *T*_p_ for P(VBTAC/NaSS)_97_ is higher than that for P(VBTAC/NaSS)_20_. The [NaCl] dependence of *T*_p_ for P(VBTAC/NaSS)_20_ is more sensitive than that for P(VBTAC/NaSS)_97_. *T*_p_ for P(VBTAC/NaSS)_20_ decreases suddenly in a narrower [NaCl] range compared to that for P(VBTAC/NaSS)_97_ (Figure 3c). Thus, the lower-molecular-weight polyampholyte is more strongly affected by [NaCl] than the higher-molecular-weight polymer.

We studied the effect of *C*_p_ on *T*_p_ (Figure 4). The temperature dependences of %*T* for aqueous P(VBTAC/NaSS)_20_ solutions containing 0.1 M NaCl were measured at different *C*_p_ values ranging from 1.0 to 5.0 g/L. *T*_p_ shifts to higher temperature with increasing *C*_p_. Furthermore, %*T* for aqueous P(VBTAC/NaSS)_97_ at [NaCl] = 1.0 M was measured as a function of temperature at different *C*_p_ values ranging from 0.5 to 3.0 g/L. *T*_p_ for P(VBTAC/NaSS)_97_ increases with increasing *C*_p_. When *C*_p_ is high, interpolymer chain interactions occur readily. Thus, to dissociate the entangled polymer chains at high *C*_p_ requires much more energy than at low *C*_p_. Therefore, higher *C*_p_ solutions present higher *T*_p_ values. These observations indicate that *T*_p_ depends on the polymer concentration like many other UCST polymers.

We performed phase contrast optical microscopy observation for aqueous 0.1 M NaCl P(VBTAC/NaSS)_20_ and 1.0 M NaCl P(VBTAC/NaSS)_97_ solutions at *C*_p_ = 2.0 g/L (Figure 5). These aqueous solutions were observed to be turbid because the observations were performed at 20 °C, which is lower than the UCST values for each polymer. Hydrated polymer aggregates are observed. The average particle sizes for P(VBTAC/NaSS)_20_ and P(VBTAC/NaSS)_97_ as estimated from optical microscopy observations are 2.2 and 3.7 μm, respectively.

DLS measurements were performed for aqueous 0.1 M NaCl P(VBTAC/NaSS)_20_ and 1.0 M NaCl aqueous P(VBTAC/NaSS)_97_ solutions at *C*_p_ = 2.0 g/L and 70 °C to obtain *R*_h_ values and distribution data (Figure 6). Both solutions were clear because the DLS measurements were performed at 70 °C, which is higher than the UCST values of the polymers. The *R*_h_ distributions are unimodal. The *R*_h_ values for P(VBTAC/NaSS)_20_ and P(VBTAC/NaSS)_97_ are 2.4 and 3.7 nm, respectively. The polydispersity index (PDI) values for P(VBTAC/NaSS)_20_ and P(VBTAC/NaSS)_97_ are 0.12 and 0.21, respectively. These small *R*_h_ values indicate that the polymers can dissolve as unimer states in aqueous solutions above their UCSTs. The zeta potential values for P(VBTAC/NaSS)_20_ and P(VBTAC/NaSS)_97_ at 70 °C are −0.32 and +0.53 mV, respectively. The zeta potential values for P(VBTAC/NaSS)*_n_* above the UCST are near to zero mV, indicating that the charges in the polyampholytes are neutralized.

The UCST behavior was also investigated in terms of the temperature dependence on light scattering measurements (Appendix A). The *T*_p_ values obtained by light scattering measurements are in good agreement with the results obtained from %*T* measurements. The values of *R*_h_ and light scattering intensity (SI) for the aqueous P(VBTAC/NaSS)*_n_* solutions were measured as a function of temperature upon cooling. *T*_p_ was defined as the temperature at which *R*_h_ and SI suddenly increase with decreasing temperature. Above *T*_p_ the *R*_h_ values of P(VBTAC/NaSS)_20_ and P(VBTAC/NaSS)_97_ are almost constant at 2.5 and 4.0 nm, respectively. Above *T*_p_, the SI values of P(VBTAC/NaSS)_20_ and P(VBTAC/NaSS)_97_ are approximately 107 and 161 Kcps, respectively. These small *R*_h_ and SI values suggest that P(VBTAC/NaSS)*_n_* exist in unimer states above *T*_p_. Below *T*_p_, P(VBTAC/NaSS)*_n_*, forms large aggregates and the solutions become cloudy. Just below *T*_p_, unimodal *R*_h_ distributions are observed. However, it is difficult to obtain *R*_h_ distributions at temperatures significantly below *T*_p_ owing to the time dependence of aggregate size. Flocculation progresses with increasing time, and the polymer aggregates precipitate (Appendix A).

To evaluate the polarities of the hydrophobic domains formed by P(VBTAC/NaSS)*_n_* below and above the UCSTs, a hydrophobic PNA fluorescence probe was used. The maximum fluorescence wavelength (*λ*_max_) for the probe depends on the polarity of the environment around the PNA molecule [39]. When the environment around the PNA molecule is hydrophobic, *λ*_max_ shifts to shorter wavelength. At 20 and 70 °C, *λ*_max_ for the PNA probe in saturated aqueous solutions containing no polymer is observed at 463 and 452 nm, respectively (Appendix A). Thus, PNA fluorescence exhibits temperature dependence in water [40]. The *λ*_max_ values in the presence of P(VBTAC/NaSS)_20_ at 20 and 70 °C are 412 and 417 nm, respectively. The *λ*_max_ values in the presence of P(VBTAC/NaSS)_97_ at 20 and 70 °C are 417 and 423 nm, respectively (Appendix A). These results indicate that hydrophobic interactions between the main polymer chains and PNA molecules below the UCST are slightly stronger than those above the UCST. Above the UCST (at 70 °C), the P(VBTAC/NaSS)*_n_* solutions are clear. The *λ*_max_ exhibits blue shifting comparing to that without the polymers at 70 °C. This observation indicates that the PNA molecules may interact with the hydrophobic main chain even at 70 °C.

When PNaSS (*M*_w_ = 70,000) is dissolved in PNA-saturated water, *λ*_max_ at 20 °C is observed at 377 nm. This observation suggests that the hydrophobic main chain of PNaSS interacts with PNA at 20 °C. At 20 °C, the hydrophobic interactions between the PNaSS homopolymer and PNA molecules are stronger than those between P(VBTAC/NaSS)*_n_* and PNA as revealed by the *λ*_max_ values for PNaSS and P(VBTAC/NaSS)*_n_* being 377 and 412–417 nm, respectively. Although the aqueous PNaSS solution is clear, the aqueous P(VBTAC/NaSS)*_n_* solutions are turbid at 20 °C. These observations suggest that electrostatic interactions may present a more important contribution to the UCST mechanism than hydrophobic interactions.

In the case of polysulfobetaine labeled with a solvatochromic fluorescent end-group, it did not reveal any local environmental effects of the problem upon passing the UCST phase transition [41]. Also solvatochromic fluorescence dyes did not reveal local environmental changes upon passing the UCST phase transition of poly(ethylene glycol)-*block*-polysulfobetaine [42]. Even when using the identical probe PNA, the UCST phase transition of a polysulfobetaine showed no effect on the emission spectrum [43]. Considering these reports, it seems probable that the observed spectroscopic shift in presence of P(VBTAC/NaSS)*_n_* is due to hydrophobic nanodomains formed by the polystyrene backbone rather than to an effect of coil-to-globule transition of the copolymers [29].

An H-D isotope effect can be observed in the UCST of P(VBTAC/NaSS)*_n_* (Appendix A). LCST [44,45] and UCST [46] are sometimes affected by D_2_O, which is the D-H isotope effect. The UCST value of P(VBTAC/NaSS)*_n_* is higher in D_2_O than that in H_2_O. The *T*_p_ values of P(VBTAC/NaSS)_20_ in D_2_O and in H_2_O are 56.5 and 46.5 °C, respectively. The *T*_p_ values of P(VBTAC/NaSS)_97_ in D_2_O and in H_2_O are 63.8 and 48.7 °C, respectively. The amplitude of atomic vibrations is lower in D_2_O than that in H_2_O because deuterium is heavier than hydrogen [47], and this helps D_2_O molecules to be more structurally organized than H_2_O molecules [48]. In polymer solutions, a hydration layer typically exists around the polymer, which is commonly known as hydrophobic hydration. The hydration layer formed in D_2_O is more organized than that in H_2_O. Therefore, it requires more energy to be disrupted in D_2_O than it does in H_2_O, leading to an increase in the UCST.

## 4. Conclusions

P(VBTAC/NaSS)*_n_* were prepared using cationic VBTAC and anionic NaSS monomers in close to stoichiometric charge balance via RAFT controlled radical polymerization. The *T*_p_ values for the P(VBTAC/NaSS)*_n_* aqueous solution increase with increasing *C*_p_ and molecular weight and with decreasing NaCl concentrations. Thus, *T*_p_ can be tuned by altering *C*_p_, molecular weight, and NaCl concentration. In D_2_O, the *T*_p_ values are higher than those in H_2_O owing to the H-D isotope effect. Furthermore, fluorescence probe measurements revealed that both electrostatic and hydrophobic interactions affect the UCST phase transition behavior.

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
