# Peer review of "Upper Critical Solution Temperature (UCST) Behavior of Polystyrene-Based Polyampholytes in Aqueous Solution"

_polymers, 2019, doi:10.3390/polym11020265_

Round 1

Reviewer 1 Report

In this work, the authors proposed synthesis and UCST characterization of P(VBTAC/NaSS)n, prepared via RAFT polymerization. The UCST of P(VBTAC/NaSS)n changed upon the concentrations of NaCl and polymer, molecular weight and solvents. The aggregation of P(VBTAC/NaSS)n was found to be the electrostatic and hydrophobic effect. The work has been fully characterized and properly articulated. I would suggest to accept the paper after taking account the following comments:

Why polymers from RAFT exhibit UCST behavior, but not for those from conventional free radical polymerization? 

Could you propose possible/potential applications with P(VBTAC/NaSS)n in terms of temperature responsiveness.

Author Response

Dear Reviewer 1

Journal: Polymers

Manuscript ID: polymers-418201

Type: Article

Title: Upper Critical Solution Temperature (UCST) Behavior of Polystyrene-based Amphoteric Random Copolymers in Aqueous Solution

Authors: Komol Kanta Sharker, Yuki Ohara, Yusuke Shigeta, Shinji Ozoe, Shin-ichi Yusa

Thank you for very useful comments. The following is the revisions that we have made and our replies to the your comments (the your comments are indicated in blue color for your convenience). The line numbers refer to those in the revised manuscript unless otherwise noted. In order for you to know which parts have been changed, the text indicates the additions with yellow highlighting in one revised manuscript, and another which doesn’t contain any highlighting marks.

In this work, the authors proposed synthesis and UCST characterization of P(VBTAC/NaSS)n, prepared via RAFT polymerization. The UCST of P(VBTAC/NaSS)n changed upon the concentrations of NaCl and polymer, molecular weight and solvents. The aggregation of P(VBTAC/NaSS)n was found to be the electrostatic and hydrophobic effect. The work has been fully characterized and properly articulated. I would suggest to accept the paper after taking account the following comments:

Why polymers from RAFT exhibit UCST behavior, but not for those from conventional free radical polymerization? 

  According to Figure 2, the molecular weight of the copolymer is very important factor to determine the UCST value. Therefore, in this study, we would like to control the structure of the polymers via RAFT polymerization.

  However, we appreciate this comment. We will prepare the copolymer via conventional free radical polymerization, and we will study thermo-responsive behavior in the near future.

Could you propose possible/potential applications with P(VBTAC/NaSS)n in terms of temperature responsiveness.

At current time we don’t have any idea of the application. However we believe that the thermo-responsive copolymers can be applied for separation, catalysis, and so on.

We are grateful to the reviewer’s comments which are very useful for improving our manuscript. I hope you will find that this manuscript is suitable for publication in Polymers.

Sincerely yours,

Dr. Shin-ichi Yusa

Reviewer 2 Report

The manuscript deals with a new polyampholyte copolymer. The copolymer is made of a - at least close to - stoichiometric mixture of strong anionic and strong cationic monomers, and exhibits a phase diagram in aqueous salt solution showing a UCST-transition.

Dealing with both charge neutral polyampholytes ("polyzwitterion mimics") as well as water-based thermo- and salt-responsive systems displaying a UCST-transition, the manuscript addresses simultaneously two thriving topics of Polymer Science. Its contents match definitely the profile of POLYMERS. The manuscript reports interesting, remarkable findings, and therefore merits a priori publication.

Nevertheless, the manuscript contains a number of shortcomings and weak points in its present form, which require correction or improvement, respectively, prior to publication. In particular, the polymer structure needs reinterpretation: on the basis of the data shown, and of the current state of knowledge, the postulated "random copolymer" structure is not justified and most probably wrong. Also, pertinent references are missing. Further, in the current form, the fragmentary background provided in combination with the somewhat superficial discussion of the findings obscure the importance of the observations made.

DETAILED COMMENTS

1) Without positive proof (not given in the manuscript) the authors cannot declare the polymers to be random copolymers. This precise structural statement is not backed by the data. In fact, it is most doubtful that the copolymers made are random copolymers. Most probably, they have a close to alternating structure (see below). As the authors have used equimolar mixtures of comonomers, the analytical data indicate that copolymerization probably takes place close to the azeotropic point of the copolymerization diagram. Hence, close to alternating copolymers with a very homogeneous compositional distribution were probably produced.

Therefore, as long as the authors do not give positive proof for a random structure (by establishing the copolymerization diagram at low conversions), the attribute "random" must be removed throughout the manuscript, i.e. in the title, abstract and text, and the structural discussion fundamentally revised.

a) The term "random copolymer" refers to the microstructure of the copolymer formed, namely, it informs on the monomer sequence distribution within the individual macromolecules. By definition (see e.g., Pure Appl. Chem. (1996) 68(12), 2287 doi 10.1351/ pac199668122287, or the Purple Book of the IUPAC), the sequence of the comonomers' constitutional repeat units (CRU) in random copolymers follows a Bernoullian statistics. In terms of copolymerization reactivity ratios, this corresponds to r1 × r2 =1 (including the simplest case of r1 = r2 =1.)

b) If an equimolar mixture of comonomers yields a 1:1 copolymer at low conversion, this implies that copolymerization is conducted in the proximity of an azeotropic point. Still, it cannot be distinguished, whether the system presents a simple azeotropic copolymerization (r1 × r2 <0 with r1 ≈ r2) yielding statistical copolymers, or the extreme cases of either an ideal azeotropic copolymerization (r1 = r2 =1) yielding random copolymers, or an alternating copolymerization (r1 × r2 =0) yielding alternating copolymers. The experimental data exclude only the cases of a simple ideal copolymerization (r1 × r2 =1 with r1 , r2 ≠1) that would lead to random copolymers, or of gradient copolymers due to a notable compositional drift. Recall also that - as the law of mass conservation applies -, for increasing conversions (here, yields were as high as 60-70%), the copolymer composition becomes increasingly "smoothened" thereby increasingly pretending that the copolymer composition mirrors at lest approximately the composition of the reaction mixture.

c) Even if employing two styrenic comonomers, in the case of differing substituents (conjugated or not, electron-withdrawing or electron-donating, etc.), as it is the case for the pair VBTAC- NaSS, it cannot be taken for granted that r1 = r2 =1, as done by the authors (lines 64-67). For instance, r-values for the pair styrene/4-vinylpyridine are r1 ≈ r2 ≈0.5. In the particular case of a controlled polymerization not conducted at the azeotropic point, such a scenario results not even in the formation of a statistical copolymer, but of a gradient one.

d) Furthermore, the copolymerization of cationic/anionic monomer pairs is not new. Even in the case of close to identical polymerizable units (as analogously substituted cationic + anionic pairs of acrylamides or of methacrylates), the copolymerization studies undertaken generally indicate a copolymerization behavior with r1 × r2 <<1. This means that alternating comonomer sequences seem generally preferred for the copolymerization of cationic with anionic monomers, independent of the polymerizable groups involved.

See e.g.: Katchalsky and Miller, J. Polym. Sci. (1954) 13, 57;  Salamone et al., J. Polym. Sci. Part A, Polym. Chem. (1980) 18, 1983;  Salamone et al., J. Polym. Sci. Part C, Lett (1985) 23, 655;  Salamone et al., Polymer (1985) 26, 1234;  Salamone et al., J. Macromol. Sci. A - Chem. (1988) 25, 811;  McCormick and Johnson, Macromolecules (1988) 21, 686;  Hahn et al., Acta Polym. (1989) 40, 36, Corpart et al., Polymer (1993) 34, 3873.

2) Abstract, line 17: "... a stoichiometrically charge-neutralized ..."

This is an assumption, not experimentally proven. Change the wording accordingly! The integration of the purely resolved 1H NMR points to a 50-50 incorporation of the monomers, but the precision of the data is low. The comparison of Fig.1 with the values given in Table 1 indicates that the insinuated precision of the data (better than 1%) is a non-justified claim. From the spectra shown, any composition between 60-40 to 40-60 could be deduced, or even worse. More and more precise analytical data (e.g. by elemental analysis) of polymers produced at conversions <10% must be given to support this claim, or the structural description adapted to "approximately 50-50" etc. . The precise knowledge of the composition and the copolymerization parameters is crucial to decide, whether (i) truly zero-net-charge polymer chains were made, and (ii) how chemically homogeneous the copolymers are. This solid information is needed for an appropriate discussion of the observations on the solubilities and the cloud points etc. .

3) Abstract, lines 20-21: "... shifted to higher temperatures by increasing the polymer concentration ..."

Specify the polymer concentration range studied. When the UCST is reached, the cloud point will inevitably decrease with further increasing polymer concentration.

4) referencing in the introduction lines 28-74.

References are not "obligatory ornaments" of a scientific paper, but an important feature to help the reader in understanding the significance of the work reported, and for enabling a larger view on the findings. As the authors present a study on the UCST-behavior in aqueous solution, the referencing must emphasize the existing work on this scenario. However, such references are scarce, and hints to essential work are missing

(e.g., Katchalsky and Miller, J. Polym. Sci. (1954) 13, 57;  McCormick and Johnson, Macromolecules (1988) 21, 694;  Corpart and Candau, Macromolecules (1993) 26, 1333;  Yang and Jhon, J. Polym. Sci. Part A, Polym. Chem. (1996) 33, 2613;  Takeoka et al., Phys. Rev. Lett. (1999) 82, 4863;  Sun et al., Polym. Sci. Ser. C (2017) 59, 11;  Gonsior and Ritter, Macromol. Chem. Phys. (2012) 213, 382).

In contrast, most of the references given deal either with LCST behavior, or with solution phase diagrams in non-aqueous solutions, and thus are not appropriate. Recall that UCST behavior in polymer solution is normal for the "real solution" scenario, and therefore a priori trivial; only in water, UCST behavior of polymers has been exceptional up to now, while LCST behavior (rather exceptional in most solvents) is the "normal" case for non-ionic polymers (see e.g. Aseyev et al., Adv. Polym. Sci. (2011) 242, 29). In this context, refs. 8+9 are completely misleading. Moreover, the references 1-13 seem to be picked randomly out of a large pool of studies of "me, too" character, with no respect whether they really match the information to be backed up and/or are particularly helpful for the reader (i.e., representing either the first studies, seminal work, or high quality up-to-date reviews). Such a "pseudo-referencing" is useless, and a sign of poor workmanship.

5) line 39: "... far fewer UCST polymer have been reported ..."

This statement is only true for polymers in water. Specifiy in the text.

6) line 47: add ref: Monroy Soto and Galin, Polymer (1984) 25, 254.

7) line 52: ref.21 refers neither to polyampholytes made by copolymerization of cation/anion pairs nor to the use of the RAFT method. Replace!

8) line 59: It should be specified that the UCST behavior observed in ref. 22 applies only to solutions in alcohols, not in water.  (see comment 4)

9) line 68: ref. 23 does not support a random copolymerization mechanism. Reformulate or remove.

10) line 78: "...we  introduce novel amphoteric random ..."

a) Are you sure, that VBTAC/NaSS copolymers have been never reported before? At least, analogous cross-linked hydrogels have been reported that show also a thermal phase transition in water (Takeoka et al., Phys. Rev. Lett. (1999) 82, 4863;  Morisada et al., Adsorption (2008) 14, 621;  please cite these studies). Thus, the copolymers may be "new" polymers, but they are by no means "novel".

b) the copolymers are derived from strong cations and anions, i.e., they are NOT amphoteric (which would mean that the overall charge is a function of pH). Instead, they represent (strong) polyampholytes.

c) as discussed above (comment 1), proof for a random microstructure is missing. In the contrary, the copolymers are most probably not "random" ones, but statistical copolymers with a strong tendency for alternation.

11) line 85. Replace "... the isotope effect ..." by "... an H-D isotope effect...".

See also below, comment no.25

12) line 93:  The purity of NaSS is given as 89%.

Did you verify this value? What are the 11 wt% of contaminants made of? Why did you not purify the monomer? And why did you not consider the missing 11 wt% when defining the NaSS comonomer quantities employed (see quantities of NaSS given in lines 101 and 108). Does this mean, that the polymers produced are nota 100:100 (as insinuated), but a 100:89 cation-anion mixture even in the optimal case?

Clarify the facts, and discuss the possible problems deriving therefrom in the manuscript.

13)  line 102:  why was the NMR solvent D2O complemented by MeOH and not by CD3OD? I suppose, the additional, strong MeOH signal interferes with the integration of the methylammonium signal at about 3 ppm, thus rendering a quantitative analysis of the spectra complicated and less precise.

14) lines 113-115: Isolation of the copolymers.

Did you verify by elemental analysis that the copolymers are devoid of residual NaCl? Even small amounts of salt may alter the cloud points of UCST systems markedly (see, e.g., Hildebrand et al., Polym. Chem. (2017) 8, 310).

15) lines 135-137. "... composed of equimolar amounts of ..."

Relativize this statement. See comment nr.2.

16) line 147-148 discussion of the induction period.

Modify the discussion, and refer correctly to the contents of the citations 26 and 27 given. Ref.26 refers to a light triggered process, and is therefore not an appropriate "explanation" for your observation. Ref. 27 specifies the importance of matching Z and R groups for a given RAFT agent to minimize/overcome an induction period. In ref.27, acrylic monomers are employed with various dithiobenzoates, which give too stabilized radical adducts for being optimal RAFT agents for these monomers. Here, the authors use a dithiobenzoate CTA with styrenic monomers. This is a much better match of reactivity. There is a good chance that the short induction period is simply due to some contaminants (as traces of oxygen, or impurities in NaSS or VBTAC, which both were used as received) which must be consumed before polymerization can occur, but not a mechanistic particularity. Therefore, the present discussion is too superficial and should be improved.

17) Table 1.

The insinuated precision of the experimental data for conversion, VBTAC content, and molar mass cannot be correct. It is much (!) lower than written. Indicate the true precisions/error bars. 

18) line 185: incorrect term: replace "phenyl" (= -C6H5) by "aryl"

19) line 186-187: reconsider and rewrite the statement to make the reader know what is proven fact, what is assumption (and how probable), and what is merely wishful thinking. For the discussion of the (most interesting!) observations of the solution behavior it is crucial to understand correctly the copolymer structure, whether it is exactly a 1:1 copolymer (and thus indeed charge-neutralized), or whether it is only close, but nevertheless still a lightly charged polycation (or polyanion).

20) line 201, figures 2a and 2c, and line 206 ("... values from 0 to ...)

Remarkable results! Still, how can you be sure that the content of NaCl is truly 0? See comment 14.

21) line 209:  Should "ae 55.7" read "at 55.7" ?

22) line 235:  "... can be controlled by altering the polymer concentration".

So what? This is not a new, but a trivial finding. Inherently, polymer concentration is a variable in a phase diagram. Modify the statement.

23) line 274:  Replace "... are precipitated ..." by "... precipitate ..."

24) lines 282 - 288:  Spectroscopic shift observed in the presence of the copolymers.

This finding is remarkable, and merits a more profound discussion. In fact, previous studies using polyzwitterions (i.e., structurally closely related polymer systems) did not reveal any local environmental effects on solvatochromic probes upon passing the UCST phase transition (see e.g., Hildebrand et al., Polymer (2017) 122, 347;  Nizardo et al., Polymers (2018) 10, 325). Even when using the identical probe PNA, the UCST phase transition of a polyzwitterion showed no effect on the emission spectrum (Arotcarena et al., J. Am. Chem. Soc. (2002) 124, 3787). Considering these reports, it seems probable that the observed spectroscopic shift in presence of the polyampholyte is due to hydrophobic nanodomains formed by the polystyrene backbone (as known from studies of the aqueous solution behavior of styrene maleic anhydride/acid copolymers;  see e.g. Morisada et al., Adsorption (2008) 14, 621) rather than to an effect of the copolymers' coil-to-globule transition.

25) lines 297-305: discussion of the D-H isotope effect.

The finding is most interesting, and inscribes well into previous reports on similar H-D isotope effects for polymers showing a UCST-transition in water (e.g., ref.11,  or  Hou and Wu, Soft Matter (2015) 11, 7059). However, H-D isotope effects are marginal at best in the case of polymers showing an LCST-transition in water (e.g., ref.11;  Mao et al., Macromolecules (2004) 37, 1031;  Zhang et al., J. Am. Chem. Soc. (2005) 127, 14505).

Accordingly, the explanation via hydrophobic hydration - that should mainly affect an LCST transition - is too simplistic and cannot be correct.

Author Response

Dear Reviewer 2

Journal: Polymers

Manuscript ID: polymers-418201

Type: Article

Title: Upper Critical Solution Temperature (UCST) Behavior of Polystyrene-based Amphoteric Random Copolymers in Aqueous Solution

Authors: Komol Kanta Sharker, Yuki Ohara, Yusuke Shigeta, Shinji Ozoe, Shin-ichi Yusa

Thank you for very useful comments. The following is the revisions that we have made and our replies to the your comments (the your comments are indicated in blue color for your convenience). The line numbers refer to those in the revised manuscript unless otherwise noted. In order for you to know which parts have been changed, the text indicates the additions with yellow highlighting in one revised manuscript, and another which doesn’t contain any highlighting marks.

The manuscript deals with a new polyampholyte copolymer. The copolymer is made of a - at least close to - stoichiometric mixture of strong anionic and strong cationic monomers, and exhibits a phase diagram in aqueous salt solution showing a UCST-transition.

Dealing with both charge neutral polyampholytes ("polyzwitterion mimics") as well as water-based thermo- and salt-responsive systems displaying a UCST-transition, the manuscript addresses simultaneously two thriving topics of Polymer Science. Its contents match definitely the profile of POLYMERS. The manuscript reports interesting, remarkable findings, and therefore merits a priori publication.

Nevertheless, the manuscript contains a number of shortcomings and weak points in its present form, which require correction or improvement, respectively, prior to publication. In particular, the polymer structure needs reinterpretation: on the basis of the data shown, and of the current state of knowledge, the postulated "random copolymer" structure is not justified and most probably wrong. Also, pertinent references are missing. Further, in the current form, the fragmentary background provided in combination with the somewhat superficial discussion of the findings obscure the importance of the observations made.

DETAILED COMMENTS

1) Without positive proof (not given in the manuscript) the authors cannot declare the polymers to be random copolymers. This precise structural statement is not backed by the data. In fact, it is most doubtful that the copolymers made are random copolymers. Most probably, they have a close to alternating structure (see below). As the authors have used equimolar mixtures of comonomers, the analytical data indicate that copolymerization probably takes place close to the azeotropic point of the copolymerization diagram. Hence, close to alternating copolymers with a very homogeneous compositional distribution were probably produced.

Therefore, as long as the authors do not give positive proof for a random structure (by establishing the copolymerization diagram at low conversions), the attribute "random" must be removed throughout the manuscript, i.e. in the title, abstract and text, and the structural discussion fundamentally revised.

a) The term "random copolymer" refers to the microstructure of the copolymer formed, namely, it informs on the monomer sequence distribution within the individual macromolecules. By definition (see e.g., Pure Appl. Chem. (1996) 68(12), 2287 doi 10.1351/ pac199668122287, or the Purple Book of the IUPAC), the sequence of the comonomers' constitutional repeat units (CRU) in random copolymers follows a Bernoullian statistics. In terms of copolymerization reactivity ratios, this corresponds to r1 × r2 =1 (including the simplest case of r1 = r2 =1.)

Thank you very much for the useful comment. We deleted the word “random”, because now we don’t have any evidence of random copolymerization. Instead of “random copolymer”, we use “statistical copolymer” in this manuscript.

b) If an equimolar mixture of comonomers yields a 1:1 copolymer at low conversion, this implies that copolymerization is conducted in the proximity of an azeotropic point. Still, it cannot be distinguished, whether the system presents a simple azeotropic copolymerization (r1 × r2 <0 with r1 ≈ r2) yielding statistical copolymers, or the extreme cases of either an ideal azeotropic copolymerization (r1 = r2 =1) yielding random copolymers, or an alternating copolymerization (r1 × r2 =0) yielding alternating copolymers. The experimental data exclude only the cases of a simple ideal copolymerization (r1 × r2 =1 with r1 , r2 ≠1) that would lead to random copolymers, or of gradient copolymers due to a notable compositional drift. Recall also that - as the law of mass conservation applies -, for increasing conversions (here, yields were as high as 60-70%), the copolymer composition becomes increasingly "smoothened" thereby increasingly pretending that the copolymer composition mirrors at lest approximately the composition of the reaction mixture.

Unfortunately, now we did not perform to determine monomer reactivity ratios with low conversion conditions. In this study we would like to focus on UCST behavior of the copolymer. The polymerization mechanism of cationic and anionic monomers including other pairs of opposite charged monomers such as acrylamide and acrylate type monomers will be studied in the near future.

c) Even if employing two styrenic comonomers, in the case of differing substituents (conjugated or not, electron-withdrawing or electron-donating, etc.), as it is the case for the pair VBTAC- NaSS, it cannot be taken for granted that r1 = r2 =1, as done by the authors (lines 64-67). For instance, r-values for the pair styrene/4-vinylpyridine are r1 ≈ r2 ≈0.5. In the particular case of a controlled polymerization not conducted at the azeotropic point, such a scenario results not even in the formation of a statistical copolymer, but of a gradient one.

At this time, we don’t determine monomer reactivity rations of VBTAC and NaSS.

Figure R1. 1H NMR (left) and corresponding typical fitting function (right) for the DOSY measurements of (a) VBTAC monomer (b) NaSS monomer with 0.5 M monomer concentration and 0.6 M NaCl concentration and (c) equimolar mixture of both the monomers.

We performed DOSY NMR measurements (Figure R1) for VBTAC and NaSS in D2O containing 0.6 M NaCl to determine diffusion coefficient (D). The D values for VBTAC and NaSS were 3.41 × 10-10 m2/s and 3.92 × 10-10 m2/s, respectively. When the two monomers were mixed, the D value of VBTAC and NaSS were the same as 2.45 × 10-10 m2/s. These observations suggest that there are some interactions between VBTAC and NaSS, such as electrostatic, hydrophobic, charge-transfer, or π-π interactions. Therefore, now we think that r1 and r2 may not be 1. That means maybe we obtained alternative copolymer.  

In this study we would like to focus on UCST behavior of the copolymer. The polymerization mechanism of cationic and anionic monomers including other pairs of opposite charged monomers such as acrylamide and acrylate type monomers will be studied in the near future.

d) Furthermore, the copolymerization of cationic/anionic monomer pairs is not new. Even in the case of close to identical polymerizable units (as analogously substituted cationic + anionic pairs of acrylamides or of methacrylates), the copolymerization studies undertaken generally indicate a copolymerization behavior with r1 × r2 <<1. This means that alternating comonomer sequences seem generally preferred for the copolymerization of cationic with anionic monomers, independent of the polymerizable groups involved.

See e.g.: Katchalsky and Miller, J. Polym. Sci. (1954) 13, 57;  Salamone et al., J. Polym. Sci. Part A, Polym. Chem. (1980) 18, 1983;  Salamone et al., J. Polym. Sci. Part C, Lett (1985) 23, 655;  Salamone et al., Polymer (1985) 26, 1234;  Salamone et al., J. Macromol. Sci. A - Chem. (1988) 25, 811;  McCormick and Johnson, Macromolecules (1988) 21, 686;  Hahn et al., Acta Polym. (1989) 40, 36, Corpart et al., Polymer (1993) 34, 3873.

We have added the following description on Line 154-456, and the suggested reference papers:

  Generally, alternating comonomer sequences seem generally preferred for the copolymerization of cationic with anionic monomers, independent of the polymerizable groups involved.

20. Katchalsky, A.; Shavit, N.; Eisenberg, H. Dissociation of weak polymeric acids and bases. J. Poly. Sci.1954, 3, 69–84.

34. Salamone, J.C.; Tsai, C.C.; Olson, A.P.; Watterson, A.C. Ampholytic polystyrene ionomers from cationic–anionic monomer pairs. J. Polym. Sci. Part A 1980, 18, 2983–2992.

35. Salamone, J.C.; Raheja, M.K.; Anwaruddin, Q.; Watterson, A.C. Polymerization of vinylpyridinium salts. XIII. Preparation of 4‐vinyl‐N‐methylpyridinium p‐styrenesulfonate charge transfer ion‐pair comonomer. J. Polym. Sci. Polym. Let. Ed. 1985, 23, 655–659.

36. Salamone, J.C.; Ahmed, I.; Rodriguez, E.L.; Quach, L.; Watterson, A.C. Synthesis and solution properties of ampholytic acrylamide ionomers. J. Macromol. Sci. Chem. 1988, A25, 811–837.

37. McCormick, C.L.; Johnson, C.B. Water-soluble polymers. 28. Ampholytic copolymers of sodium 2-acrylamido-2-methylpropanesulfonate with (2-acrylamido-2-methylpropyl)dimethylammonium chloride: synthesis and characterization. Macromolecules 1988, 21, 686–693.

38. Corpart, J.M.; Selb, J.; Candau, F. Characterization of high charge density ampholytic copolymers prepared by microemulsion polymerization. Polymer 1993, 34, 3873–3886.

Unfortunately, we cannot read Germany. Therefore, we don’t cite M. Hahn, J. Kötz, K.‐J. Linow, B. Philipp, Synthese, Charakterisierung und Symplexbildung von Polyampholyten aus ungesättigten Discarbonsäure‐ und Allylaminderivaten. 1. Mitteilung: Synthese und Charakterisierung von Copolymeren aus Maleinsäre und Allylaminderivaten. Acta Polymeriza, 1989, 40, 36-43.

2) Abstract, line 17: "... a stoichiometrically charge-neutralized ..."

This is an assumption, not experimentally proven. Change the wording accordingly! The integration of the purely resolved 1H NMR points to a 50-50 incorporation of the monomers, but the precision of the data is low. The comparison of Fig.1 with the values given in Table 1 indicates that the insinuated precision of the data (better than 1%) is a non-justified claim. From the spectra shown, any composition between 60-40 to 40-60 could be deduced, or even worse. More and more precise analytical data (e.g. by elemental analysis) of polymers produced at conversions <10% must be given to support this claim, or the structural description adapted to "approximately 50-50" etc. . The precise knowledge of the composition and the copolymerization parameters is crucial to decide, whether (i) truly zero-net-charge polymer chains were made, and (ii) how chemically homogeneous the copolymers are. This solid information is needed for an appropriate discussion of the observations on the solubilities and the cloud points etc. .

Thank you very much for useful comments. We have re-measure 1H NMR to determine the true content of VBTAC in the copolymer. We have three times measured the same sample, and then calculate the average value. The results were summarized in Table 1. The VBTAC contents in P(VBTAC/NaSS)20 and P(VBTAC/NaSS)97 were 48.2 ± 0.6 and 52.3 ± 1.0 mol%. In Abstract we used about 50 mol%.

3) Abstract, lines 20-21: "... shifted to higher temperatures by increasing the polymer concentration ..."

Specify the polymer concentration range studied. When the UCST is reached, the cloud point will inevitably decrease with further increasing polymer concentration.

We have indicated the concentration range on Line 22-23.

4) referencing in the introduction lines 28-74.

References are not "obligatory ornaments" of a scientific paper, but an important feature to help the reader in understanding the significance of the work reported, and for enabling a larger view on the findings. As the authors present a study on the UCST-behavior in aqueous solution, the referencing must emphasize the existing work on this scenario. However, such references are scarce, and hints to essential work are missing

(e.g., Katchalsky and Miller, J. Polym. Sci. (1954) 13, 57;  McCormick and Johnson, Macromolecules (1988) 21, 694;  Corpart and Candau, Macromolecules (1993) 26, 1333;  Yang and Jhon, J. Polym. Sci. Part A, Polym. Chem. (1996) 33, 2613;  Takeoka et al., Phys. Rev. Lett. (1999) 82, 4863;  Sun et al., Polym. Sci. Ser. C (2017) 59, 11;  Gonsior and Ritter, Macromol. Chem. Phys. (2012) 213, 382).

We have added the suggested reference papers:

20. Katchalsky, A.; Shavit, N.; Eisenberg, H. Dissociation of weak polymeric acids and bases. J. Poly. Sci.1954, 3, 69–84.

21. McCormick, C.L.; Johnson, C.B. Water-soluble copolymers. 29. Ampholytic copolymers of sodium 2-acrylamido-2-methylpropanesulfonate with (2-acrylamido-2-methylpropyl)dimethylammonium chloride: solution properties. Macromolecules 1988, 21, 694–699.

22. Corpart, J.M.; Candan, F. Aqueous solution properties of ampholytic copolymers prepared in microemulsions. Macromolecules 1993, 26, 1333–1343.

23. Yang, J.H.; John, M.S. The conformation and dynamics study of amphoteric copolymers, P(sodium 2‐methacryloyloxyethanesulfonate‐co‐2‐methacryloyloxyethyltrimethylammonium iodide), using viscometry, 14N‐, and 23Na‐NMR. J. Polym. Sci. Part A 1995, 33, 2613–2621.

24. Takeoka, Y.; Berker, A.N.; Du, R.; Enoki, T.; Grosberg, A.; Kardar, M.; Oya, T.; Tanaka, K.; Wang, G.; Yu, X.; Tanaka, T. First Order Phase Transition and Evidence for Frustrations in Polyampholytic Gels. Phys. Rev. Lett. 1999, 82, 4863–4865.

25. Sun, T.L.; CuiJian, K.; Gong, P. Tough, self-recovery and self-healing polyampholyte hydrogels. Polym. Sci., Ser. C 2017, 59, 11–17.

In contrast, most of the references given deal either with LCST behavior, or with solution phase diagrams in non-aqueous solutions, and thus are not appropriate. Recall that UCST behavior in polymer solution is normal for the "real solution" scenario, and therefore a priori trivial; only in water, UCST behavior of polymers has been exceptional up to now, while LCST behavior (rather exceptional in most solvents) is the "normal" case for non-ionic polymers (see e.g. Aseyev et al., Adv. Polym. Sci. (2011) 242, 29). In this context, refs. 8+9 are completely misleading. Moreover, the references 1-13 seem to be picked randomly out of a large pool of studies of "me, too" character, with no respect whether they really match the information to be backed up and/or are particularly helpful for the reader (i.e., representing either the first studies, seminal work, or high quality up-to-date reviews). Such a "pseudo-referencing" is useless, and a sign of poor workmanship.

We have selected the first studies, seminal work, or high quality up-to-date reviews as references.

We have changed the references:

1. Afrassiabi, A.; Hoffman, A.S.; Cadwell, L.A. Effect of temperature on the release rate of biomolecules from thermally reversible hydrogels. J. Membrane Sci. 1987, 33, 191–200.

2. Wei, H.; Cheng, S.X.; Zhang, X.Z.; Zhuo, R.X. Thermo-sensitive polymeric micelles based on poly(N-isopropylacrylamide) as drug carriers. Prog. Polym. Sci. 2009, 34, 893–910.

3. Kurisawa, M.; Yokoyama, M.; Okano, T. Gene expression control by temperature with thermo-responsive polymeric gene carriers. J. Controlled Release 2000, 69, 127–137.

4. Abdelaty M.S.A. Environmental functional photo-cross-linked hydrogel bilayer thin films from vanillin (part 2): temperature-responsive layer A, functional, temperature and pH layer B. Polym. Bull. 2018, 75, 4837–4858.

6. Jańczewski, D.; Tomczak, N.; Han, M.Y. Vancso, G.J. Stimulus responsive PNIPAM/QD hybrid microspheres by copolymerization with surface engineered QDs. Macromolecules200942, 1801–1804.

7. Pan, J.; Zhang, L.; Bai, L.; Zhang, Z.; Chen, H.; Cheng, Z.; Zhu, X. Atom transfer radical polymerization of methyl methacrylate with a thermo-responsive ligand: construction of thermoregulated phase-transfer catalysis in an aqueous–organic biphasic system. Polym. Chem. 2013, 4, 2876–2883.

8. Aseyev, V.; Tenhu, H.; Winnik, F.M. Non-ionic thermoresponsive polymers in water. Adv. Polym. Sci. 2011, 242, 29–89.

9. Seuring, J.; Agarwal, S. Polymers with upper critical solution temperature in aqueous solution. Macromol. Rapid Commun. 2012, 33, 1898–1920.

11. Halperin, A.; Kroeger, M.; Winnik, F.M. Poly(N-isopropylacrylamide) phase diagrams: Fifty years of research. Angew. Chem. Int. Edit 2015, 54, 15342–15367.

12. Meeussen, F.; Nies, E.; Berghmans, H.; Verbrugghe, S.; Goethals, E.; Du Prez, F. Phase behavior of poly(N-vinyl caprolactam) in water. Polymer 2000, 41, 8597–8602.

13. Vancoillie, G.; Frank, D.; Hoogenboom, R. Thermoresponsive poly(oligo ethylene glycol acrylates). Prog. Polym. Sci. 2014, 39, 1074–1095.

5) line 39: "... far fewer UCST polymer have been reported ..."

This statement is only true for polymers in water. Specifiy in the text.

We have described that can be observed in water on Line 41.

6) line 47: add ref: Monroy Soto and Galin, Polymer (1984) 25, 254.

We have added the following reference:

19. Monroy Soto, V.M.; Galin, J.C. Poly(sulphopropylbetaines): 2. Dilute solution properties. Polymer 1984, 25, 254–262.

7) line 52: ref.21 refers neither to polyampholytes made by copolymerization of cation/anion pairs nor to the use of the RAFT method. Replace!

We have replaced the reference paper to the following:

27. Wang, R.; Lowe, A.B. RAFT polymerization of styrenic-based phosphonium monomers and a new family of well-defined statistical and block polyampholytes. J. Polym. Sci. Part A 2007, 45, 2468–2483.

8) line 59: It should be specified that the UCST behavior observed in ref. 22 applies only to solutions in alcohols, not in water.  (see comment 4)

We have indicated that the UCST behavior can be observed in alcohols.

This was described on Line 63.

9) line 68: ref. 23 does not support a random copolymerization mechanism. Reformulate or remove.

We have deleted ref. 23.

10) line 78: "...we  introduce novel amphoteric random ..."

a) Are you sure, that VBTAC/NaSS copolymers have been never reported before? At least, analogous cross-linked hydrogels have been reported that show also a thermal phase transition in water (Takeoka et al., Phys. Rev. Lett. (1999) 82, 4863;  Morisada et al., Adsorption (2008) 14, 621;  please cite these studies). Thus, the copolymers may be "new" polymers, but they are by no means "novel".

We have deleted “novel”.

We have cited the following related reference papers:

24. Takeoka, Y.; Berker, A.N.; Du, R.; Enoki, T.; Grosberg, A.; Kardar, M.; Oya, T.; Tanaka, K.; Wang, G.; Yu, X.; Tanaka, T. First Order Phase Transition and Evidence for Frustrations in Polyampholytic Gels. Phys. Rev. Lett. 1999, 82, 4863–4865.

29. Morisada, S.; Suzuki, H.; Emura, S.; Hirokawa, Y.; Nakano, Y. Temperature-swing adsorption of aromatic compounds in water using polyampholyte gel. Adsorption 2008, 14, 621–628.

b) the copolymers are derived from strong cations and anions, i.e., they are NOT amphoteric (which would mean that the overall charge is a function of pH). Instead, they represent (strong) polyampholytes.

Thank you very much for the appropriate comment. Indeed we confused “amphoteric” and “polyampholytes”. We have replaced all “amphoteric polymer” to “polyampholyte”.

c) as discussed above (comment 1), proof for a random microstructure is missing. In the contrary, the copolymers are most probably not "random" ones, but statistical copolymers with a strong tendency for alternation.

We have changed the word from “random” to “statistical”.

Honestly say, at this time we don’t have evidence that the polymer is alternative or random.

11) line 85. Replace "... the isotope effect ..." by "... an H-D isotope effect...".

See also below, comment no.25

We have replaced from "... the isotope effect ..." to "... an H-D isotope effect...".

12) line 93:  The purity of NaSS is given as 89%.

Did you verify this value? What are the 11 wt% of contaminants made of? Why did you not purify the monomer? And why did you not consider the missing 11 wt% when defining the NaSS comonomer quantities employed (see quantities of NaSS given in lines 101 and 108). Does this mean, that the polymers produced are nota 100:100 (as insinuated), but a 100:89 cation-anion mixture even in the optimal case?

Clarify the facts, and discuss the possible problems deriving therefrom in the manuscript.

We have wrong to indicate the monomer, NaSS. In fact we have use NaSS purchased from Tokyo Chemical Industry (98.0% estimated by HPLC). We have revise the description on Line 90-91.

13)  line 102:  why was the NMR solvent D2O complemented by MeOH and not by CD3OD? I suppose, the additional, strong MeOH signal interferes with the integration of the methylammonium signal at about 3 ppm, thus rendering a quantitative analysis of the spectra complicated and less precise.

We have not used CD3OD, because methanol did not disturb to estimate the conversion from 1H NMR (Figure R2). We have used aryl and vinyl protons to estimate the conversion.

Figure R2. 1H NMR spectra of P(VBTAC/NaSS)97 in D2O; (a) before and (b) after polymerization at 70 °C.

14) lines 113-115: Isolation of the copolymers.

Did you verify by elemental analysis that the copolymers are devoid of residual NaCl? Even small amounts of salt may alter the cloud points of UCST systems markedly (see, e.g., Hildebrand et al., Polym. Chem. (2017) 8, 310).

We have not performed elemental analysis. However, we have performed dialysis method to purify the polymers. Therefore, we believe that the polymer does not contain excess NaCl. The about 20 mL polymer aqueous solution was dialyzed against about 10 L pure water, and the water was changed two times a day. We have checked the presence of NaCl using conductivity. Only 1.2 M NaCl aqueous solution was dialyzed against pure water using the same method of purification of the copolymers. The conductivity of pure water and 1.2 M NaCl were 1.07 1.07 μS/cm and more than 20mS/cm, respectively. After dialysis the conductivity of the solution was 1.10 μS/cm.

15) lines 135-137. "... composed of equimolar amounts of ..."

Relativize this statement. See comment nr.2.

We have deleted the description.

16) line 147-148 discussion of the induction period.

Modify the discussion, and refer correctly to the contents of the citations 26 and 27 given. Ref.26 refers to a light triggered process, and is therefore not an appropriate "explanation" for your observation. Ref. 27 specifies the importance of matching Z and R groups for a given RAFT agent to minimize/overcome an induction period. In ref.27, acrylic monomers are employed with various dithiobenzoates, which give too stabilized radical adducts for being optimal RAFT agents for these monomers. Here, the authors use a dithiobenzoate CTA with styrenic monomers. This is a much better match of reactivity. There is a good chance that the short induction period is simply due to some contaminants (as traces of oxygen, or impurities in NaSS or VBTAC, which both were used as received) which must be consumed before polymerization can occur, but not a mechanistic particularity. Therefore, the present discussion is too superficial and should be improved.

We have deleted reference 26. Instead, we have added a reference paper (33. Barner‐Kowollik, C.; Buback, M.; Charleux, B.; Coote, M.L.; Drache, M.; Fukuda, T.;  Goto, A.; Klumperman, B.; Lowe, A.B.; Mcleary, J.B.; Moad, G.; Monteiro, M.J.;  Sanderson, R.D.; Tonge, M.P.; Vana, P. Mechanism and kinetics of dithiobenzoate‐mediated RAFT polymerization. I. The current situation. J. Polym. Sci. Part A 2006, 44, 58095831.) which described induction period of styrene polymerization using a ditiobenzoate type chain transfer agent. Furthermore, we have added discussion on the possibility of impurity on Line 144-145.

Sometimes induction period can be observed due to contaminations such as trace of oxygen, or impurities in monomers.

17) Table 1.

The insinuated precision of the experimental data for conversion, VBTAC content, and molar mass cannot be correct. It is much (!) lower than written. Indicate the true precisions/error bars.

We have measured 1H NMR three times for one sample. The contents of VBTAC have been changed to the average value of the three times measurements. We have added error bars in Table 1. The VBTAC contents in P(VBTAC/NaSS)20 and P(VBTAC/NaSS)97 were 48.2 ± 0.6 and 52.3 ± 1.0 mol%, respectively.

18) line 185: incorrect term: replace "phenyl" (= -C6H5) by "aryl"

We have rephrased from “phenyl” to “aryl”.

19) line 186-187: reconsider and rewrite the statement to make the reader know what is proven fact, what is assumption (and how probable), and what is merely wishful thinking. For the discussion of the (most interesting!) observations of the solution behavior it is crucial to understand correctly the copolymer structure, whether it is exactly a 1:1 copolymer (and thus indeed charge-neutralized), or whether it is only close, but nevertheless still a lightly charged polycation (or polyanion).

We have added the following description on Lines 183-190.

  However, there is no evidence of the randomness of P(VBTAC/NaSS)n, because we cannot control the sequence of VBTAC and NaSS in the copolymerization. If VBTAC and NaSS homo polymers are mixed in D2O containing 1.2 M NaCl, we may obtain the same 1H NMR spectra in Figure 2. 1H NMR data merely indicated that P(VBTAC/NaSS)n contains equimolar amounts of the VBTAC and NaSS units in the solution. The charges were nearly canceled in this solution. Indeed, zeta-potential of the aqueous solutions of P(VBTAC/NaSS)n were near to zero, which is to be discussed later. It is considered that we have obtained copolymers of VBTAC and NaSS, because both the monomers have styrene-type structures.

20) line 201, figures 2a and 2c, and line 206 ("... values from 0 to ...)

Remarkable results! Still, how can you be sure that the content of NaCl is truly 0? See comment 14.

The polymers were purified by dialysis method against pure water, and after that we obtained the polymer powder by a freeze-drying method. Therefore, we believe that P(VBTAC/NaSS)n does not contain excess NaCl. 

21) line 209:  Should "ae 55.7" read "at 55.7" ?

We have collected. "ae 55.7" was changed to "are 55.7".

22) line 235:  "... can be controlled by altering the polymer concentration".

So what? This is not a new, but a trivial finding. Inherently, polymer concentration is a variable in a phase diagram. Modify the statement.

Of course, this finding is not new. This statement is only the results of the experiments. We have changed the statement as below:

"... can be controlled by altering the polymer concentration like many other UCST polymers".

23) line 274:  Replace "... are precipitated ..." by "... precipitate ..."

We have replaced.

24) lines 282 - 288:  Spectroscopic shift observed in the presence of the copolymers.

This finding is remarkable, and merits a more profound discussion. In fact, previous studies using polyzwitterions (i.e., structurally closely related polymer systems) did not reveal any local environmental effects on solvatochromic probes upon passing the UCST phase transition (see e.g., Hildebrand et al., Polymer (2017) 122, 347;  Nizardo et al., Polymers (2018) 10, 325). Even when using the identical probe PNA, the UCST phase transition of a polyzwitterion showed no effect on the emission spectrum (Arotcarena et al., J. Am. Chem. Soc. (2002) 124, 3787). Considering these reports, it seems probable that the observed spectroscopic shift in presence of the polyampholyte is due to hydrophobic nanodomains formed by the polystyrene backbone (as known from studies of the aqueous solution behavior of styrene maleic anhydride/acid copolymers;  see e.g. Morisada et al., Adsorption (2008) 14, 621) rather than to an effect of the copolymers' coil-to-globule transition.

Thank you very much very useful comments. Based on significant comments from the reviewer, we added the following sentences on lines 301-308 and references.

  In the case of polysulfobetaine labeled with a solvatochromic fluorescent end-group did not reveal any local environmental effects of the proble upon passing the UCST phase transition [41. Hildebrand, V.; Heydenreich, M.; Laschewsky, A.; Moller, H.M.; Müller-Buschbaum, P.; Papadakis, C.M.; Schanzenbach, D.; Wischerhoff, E. “Schizophrenic” self-assembly of dual thermoresponsive block copolymers bearing a zwitterionic and a non-ionic hydrophilic block. Polymer 2017, 122, 347357.]. Also solvatochromic fluorescence dyes did not reveal local environmental changes upon passing the UCST phase transition of poly(ethylene glycol)-block-polysulfobetaine [42. Nizardo, N.M.; Schanzenbach, D.; Schönemann, E.; Laschewsky, A. Exploring poly(ethylene glycol)-polyzwitterion diblock copolymers as biocompatible smart macrosurfactants featuring UCST-phase behavior in normal saline solution. Polymers 2018, 10, 325.]. Even when using the identical probe PNA, the UCST phase transition of a polysulfobetaine showed no effect on the emission spectrum [43. Arotçaréna, M.; Heise, B.; Ishaya, S.; Laschewsky, A. Switching the inside and the outside of aggregates of water-soluble block copolymers with double thermoresponsivity. J. Am. Chem. Soc. 2002, 124, 3787–3793.]. Considering these reports, it seems probable that the observed spectroscopic shift in presence of P(VBTAC/NaSS)n is due to hydrophobic nanodomains formed by the polystyrene backbone rather than to an effect of coil-to-globule transition of the copolymers [29. Morisada, S.; Suzuki, H.; Emura, S.; Hirokawa, Y.; Nakano, Y. Temperature-swing adsorption of aromatic compounds in water using polyampholyte gel. Adsorption 2008, 14, 621–628.].

25) lines 297-305: discussion of the D-H isotope effect.

The finding is most interesting, and inscribes well into previous reports on similar H-D isotope effects for polymers showing a UCST-transition in water (e.g., ref.11,  or  Hou and Wu, Soft Matter (2015) 11, 7059). However, H-D isotope effects are marginal at best in the case of polymers showing an LCST-transition in water (e.g., ref.11;  Mao et al., Macromolecules (2004) 37, 1031;  Zhang et al., J. Am. Chem. Soc. (2005) 127, 14505).

We have added the following references suggested from the reviewer. In this manuscript we focus on the UCST phase transition behavior of P(VBTAC/NaSS)n in aqueous solutions. Therefore, we will study the D-H isotope effect in more detail. We have added following sentence on Lines 309-310 and References.

LCST [44. Mao, H.; Li, C.; Zhang, Y.; Furyk, S.; Cremer, P.S.; Bergbreiter, D.E. High-throughput studies of the effects of polymer structure and solution components on the phase separation of thermoresponsive polymers. Macromolecueles 2004, 37, 1031–1036. 45. Zhang, Y.; Furyk, S.; Bergbreiter, D.E.; Cremer, P.S. Specific ion effects on the water solubility of macromolecules: PNIPAM and the Hofmeister series. J. Am. Chem. Soc. 2005, 127, 14505–14510.] and UCST [46. Hou, L.; Wu, P. Understanding the UCST-type transition of P(AAm-co-AN) in H2O and D2O: dramatic effects of solvent isotopes. Soft Matter 2015, 11, 7059–7065.] are sometimes affected by D2O, which is the D-H isotope effect.

We are grateful to the reviewer’s comments which are very useful for improving our manuscript. I hope you will find that this manuscript is suitable for publication in Polymers.

Sincerely yours,

Dr. Shin-ichi Yusa

Round 2

Reviewer 2 Report

The revised manuscript has remedied the problematic issues of the first version, and is suited for publication.

Nevertheless, the authors are advised to correct the few remaining, minor points as explained below

1) Keywords:   read correctly "polyampholyte"  ("o" missing)

2) lines 69 and 70:  remove "random".  (cf. discussion in the first review; the arguments apply also for the literature report refered to)

3) line 148:   remove "generally"  (repetition of previous line)

4) Table 1:  the resolution of the signals of the 1 H NMR spectra provided in Figure 1 does not justify the pretended precision of the VBTAC content of ≤1%. The correct erorr bars are significantly higher, for sure (from my personal experience, I would estimate not better than ±5 rel.%, i.e. here ±2-3 abs%). Therefore, it is more appropriate to confine the precision simply to 2 digits, and report the VBTAC content accordingly in full numbers, i.e,, 48% and 52%. Anyhow otherwise, the statement discussed below in comment (5) does not make sense (48.2% + 0.6 % = 48.8% maximum content, thus excluding true equimolarity=50%).

5) lines 186-7:  "... indicated that P(VBTAC/NaSS)n contains equimolar amounts ..."

The wording of the conclusion is somewhat misleading. Specify more precisely the data situation, and modify the phrase, e.g., as
"... indicate that P(VBTAC/NaSS)n contains about equimolar amounts ..."

or "... suggest that P(VBTAC/NaSS)n contains equimolar amounts ..."

6) line 238:  modify "... can be controlled"  into   "... depends on ..."

Author Response

Dear Reviewer 1

Journal: Polymers

Manuscript ID: polymers-418201

Type: Article

Title: Upper Critical Solution Temperature (UCST) Behavior of Polystyrene-based Polyampholytes in Aqueous Solution

Authors: Komol Kanta Sharker, Yuki Ohara, Yusuke Shigeta, Shinji Ozoe, Shin-ichi Yusa

Thank you for helpful comments. The following is the revisions that we have made and our replies to your comments (your comments are indicated in blue color for your convenience).

The revised manuscript has remedied the problematic issues of the first version, and is suited for publication.

Nevertheless, the authors are advised to correct the few remaining, minor points as explained below

1) Keywords:   read correctly "polyampholyte"  ("o" missing)

We have revised the keyword.

2) lines 69 and 70:  remove "random".  (cf. discussion in the first review; the arguments apply also for the literature report refered to)

We have deleted the word, “random”.

3) line 148:   remove "generally"  (repetition of previous line)

We have deleted the word, “generally”.

4) Table 1:  the resolution of the signals of the 1 H NMR spectra provided in Figure 1 does not justify the pretended precision of the VBTAC content of ≤1%. The correct erorr bars are significantly higher, for sure (from my personal experience, I would estimate not better than ±5 rel.%, i.e. here ±2-3 abs%). Therefore, it is more appropriate to confine the precision simply to 2 digits, and report the VBTAC content accordingly in full numbers, i.e,, 48% and 52%. Anyhow otherwise, the statement discussed below in comment (5) does not make sense (48.2% + 0.6 % = 48.8% maximum content, thus excluding true equimolarity=50%).

We have changed that VBTAC contents were 2 digit.

5) lines 186-7:  "... indicated that P(VBTAC/NaSS)n contains equimolar amounts ..."

The wording of the conclusion is somewhat misleading. Specify more precisely the data situation, and modify the phrase, e.g., as

"... indicate that P(VBTAC/NaSS)n contains about equimolar amounts ..."

or "... suggest that P(VBTAC/NaSS)n contains equimolar amounts ..."

 We have changed the phrase to “... indicate that P(VBTAC/NaSS)n contains about equimolar amounts ...”.

6) line 238:  modify "... can be controlled"  into   "... depends on ..."

We have changed the phrase “Tp depends on the polymer concentration….”.

We are grateful to the reviewer’s comments which are very useful for improving our manuscript. I hope you will find that this manuscript is suitable for publication in Polymers.

Sincerely yours,

Dr. Shin-ichi Yusa